# Screening and Characteristics Analysis of Polysaccharides from Orah Mandarin (*Citrus reticulata* cv. Orah)

**DOI:** 10.3390/foods13010082

**Published:** 2023-12-25

**Authors:** Guoming Liu, Ping Wei, Yayuan Tang, Jiemin Li, Ping Yi, Zhonglin Deng, Xuemei He, Dongning Ling, Jian Sun, Lan Zhang

**Affiliations:** 1Guangxi Academy of Agricultural Sciences, 174 East Daxue Road, Nanning 530007, China; guoming-liu@gxaas.net; 2Guangxi Key Laboratory of Fruits and Vegetables Storage-Processing Technology, 174 East Daxue Road, Nanning 530007, China; weiping@gxaas.net (P.W.); tangyayuan@gxaas.net (Y.T.); pingyi@gxaas.net (P.Y.); dzl839945664@gxaas.net (Z.D.); xuemeihegx@gxaas.net (X.H.); lingdongning@gxaas.net (D.L.); zhanglan2023@gxaas.net (L.Z.); 3Agro-Food Science and Technology Research Institute, Guangxi Academy of Agricultural Sciences, 174 East Daxue Road, Nanning 530007, China

**Keywords:** orah mandarin peel, polysaccharide, screening, characteristics analysis, NK cell activity

## Abstract

This study aimed to screen out polysaccharides with the ability to activate NK cells. Ten polysaccharides (OP) were isolated from orah mandarin (*Citrus reticulata* cv. Orah) peel using hot-water extraction combined with the alcohol precipitation method and the ultrafiltration-membrane separation method. After measuring the effects of 10 OPs on NK-92MI cell proliferation and cytotoxicity, it was found that the polysaccharide OP5 had the highest activity in vitro. OP5 can significantly promote the proliferation of and increase the gene expression of perforin, granzyme B and IFN-γ in NK-92MI cells. Its molecular weight was between 50 and 70 kDa. The identification results of monosaccharide composition indicated that OP5 was composed of arabinose (31.52%), galacturonic acid (22.35%), galactose (16.72%), glucose (15.95%), mannose (7.67%), rhamnose (2.39%), fucose (1.41%), xylose (1.30%), glucuronic acid (0.42%) and ribose (0.27%). The sugar ring of the β-configuration was the main, and that of the α-configuration was the auxiliary. These results would provide a foundation for the functional product development of OPs.

## 1. Introduction

Citrus is one of the most popular fruits worldwide. Its global marketing has exceeded 29 million tons per year. As a major producer in the world, the production of China (20 million tons) is much higher than that of the European Union (2.9 million tons), Japan (1.1 million tons), Morocco (1 million tons) and Turkey (1 million tons) [1]. Guangxi is the largest citrus production province in China, with nine varieties of citrus grown. Among them, the orah mandarin is the most popular variety of citrus among consumers. Approximately 1.2 million tons of orah mandarins are produced annually in Guangxi. With the development of the orah mandarin processing industry, a large number of peels have been produced. However, most of them are discarded commonly as waste. This has been recognized as an ecological burden for the society.

Citrus peel is rich in natural active substances such as polysaccharides, limonene, pigment and flavonoids. Among them, polysaccharides can account for 20–30% of the dry peel. Therefore, citrus peel acts as the main raw material for obtaining active polysaccharides such as pectin. The polysaccharides from citrus peel have been applied widely in the food, pharmaceutical and cosmetics industries based on their different characteristics [2]. In recent years, more and more researchers have focused on citrus polysaccharides, including extraction and separation methods [3,4], structural identification [5,6], rheological properties [4,7], antioxidant activity [6,8] and immunological activity [9,10,11,12,13].

Under the backdrop of increasing emphasis on enhancing the body’s immunity to maintain health through the intake of functional foods or health products, studying the immune-stimulating activity of polysaccharides has become one of the most important hotspots. Many polysaccharides have been proven to improve the immunity of animals and have enormous advantages and potential in the development of functional health products. For example, some polysaccharides can enhance the body’s killing effect against invading viruses by activating nature killer (NK) cells [14,15,16,17,18], but their mechanism is not fully understood. Nowadays, what people know is that the activation of NK cells is regulated by various mechanisms, including the expression and signaling of activated and inhibitory receptors on the surface of NK cells, the stimulation of cytokines such as interleukin-2 (IL-2) and interleukin-15 (IL-15), and the regulation of transcription factors related to activation and killing functions of NK cells [19]. Among them, the study of receptors is a research hotspot. There are various receptors on the surface of NK cells, such as complement receptor 3 (CR3), natural-killer group 2 member D (NKG2D), killer cell immunoglobulin-like receptors (KIR), and so on [20]. The dynamic integration of signals derived from them determines the powerful effector functions of NK cells, such as cytotoxicity and cytokine production [21,22]. After the binding of polysaccharides to receptors, NK cells are activated when the expression of activated receptors dominates. Then, the killing factors mainly composed of perforin and granzyme B are secreted to kill target cells directly. At first, the pores on the target cell membrane are made by perforin, and then granzyme B enters through the pores to activate related proteins to induce apoptosis. In addition, when the NK cells are activated, other cytokines such as interferon-γ (IFN-γ) and tumor necrosis factor-α (TNF-α) are also secreted to kill target cells indirectly by regulating the function of other immune cells [23].

Every year, a huge amount of citrus peel is produced. This is a huge source of polysaccharides. Unfortunately, few people have studied whether polysaccharides from orah mandarin peel have the ability to activate NK cells and whether they have the potential to be developed as functional foods. Accordingly, the purpose of this study was to screen out the polysaccharides of orah mandarin peel by activating NK cells through hot-water extraction, alcohol precipitation, ultrafiltration-membrane separation and activity identification. The research results will provide reference data for the high-value utilization and product development of orah mandarin peel in further research.

## 2. Materials and Methods

### 2.1. Materials and Reagents

Orah mandarins were bought from Guangxi Hyperion International Agricultural Logistics Co., Ltd. (Nanning, China). HPLC-grade mannose, ribose, rhamnose, glucuronic acid, galacturonic acid, glucose, galactose, xylose, arabinose, fucose, *N*-acetyl glucosamine and *N*-acetyl galactosamine were provided by Sigma-Aldrich Co., Ltd. (St. Louis, MO, USA). NK-92MI cells (natural killer cells in patients with human malignant non-Hodgkin’s lymphoma), Calu-1 cells (human lung cancer cells), MEMα medium, and McCoy’s 5A medium were purchased from Procell Life Science & Technology Co., Ltd. (Wuhan, China). TRIzol Reagent was obtained from Thermo Fisher Scientific (Waltham, MA, USA). HiScript II QRT SuperMix for qPCR (+gDNA wiper) and ChamQ Universal SYBR qPCR Master Mix were acquired from Vazyme Biotech Co., Ltd. (Nanjing, China). All compounds were analytical grade unless otherwise specified.

### 2.2. Preparation of Polysaccharides

#### 2.2.1. Extraction of Polysaccharides

Orah mandarin peel (10,200 g) was washed with distilled water and then dried in an oven (Taisite Instrument Co., Tianjin, China) at 40 °C. The dried peel (2100 g) was crushed into powder using an FW 177 high-speed rotating disintegrator (Taisite Instrument Co., Tianjin, China) with a speed of 24,000 rpm. The powder (100 g) was extracted with distilled water (1900 g) at 100 °C for 3 h. The supernatant was concentrated using a rotary evaporator (Shanghai Yarong Biochemical Instrument Factory, Shanghai, China) and then mixed with anhydrous ethanol in a ratio of 1:9 (*v*/*v*) at 4 °C overnight. The precipitate was collected and named as orah mandarin peel polysaccharide (OP).

#### 2.2.2. Isolation of Polysaccharides

The OP (60 g) was dissolved in distilled water (1700 mL) at 100 °C to obtain a polysaccharide solution. The solution (550 mL) was divided into 11 portions using a MinimatePall ultrafiltration system (Guangzhou Ewell Bio-Technology Co., Guangzhou, China) with 1000, 500, 300, 100, 70, 50, 30, 10, 5 and 3 kDa ultrafiltration membranes. Ten polysaccharide solutions (50 mL) were chosen to mix with absolute ethanol (450 mL), respectively, and preserved at 4 °C for 24 h. The precipitates were collected for freeze-drying to obtain the OPs with molecular weights of 3–5 kDa (OP1), 5–10 kDa (OP2), 10–30 kDa (OP3), 30–50 kDa (OP4), 50–70 kDa (OP5), 70–100 kDa (OP6), 100–300 kDa (OP7), 300–500 kDa (OP8), 500–1000 kDa (OP9) and >1000 kDa (OP10).

### 2.3. Cell Culture

#### 2.3.1. NK-92MI Cell

NK-92MI cells were cultured in MEMα medium with 0.2 mM inositol, 0.1 mM β-mercaptoethanol, 0.02 mM folic acid, 12.5% horse serum, 12.5% fetal bovine serum and 1% penicillin-streptomycin, and maintained in a 5% CO_2_ incubator (Thermo Fisher Scientific Co., Waltham, MA, USA) at 37 °C. The cells were passaged every two or three days.

#### 2.3.2. Calu-1 Cell

Calu-1 cells were cultured in McCoy’s 5A medium containing 10% fetal bovine serum and 1% penicillin-streptomycin and maintained in a 5% CO_2_ incubator (Thermo Fisher Scientific Co., Waltham, MA, USA) at 37 °C. The cells were passaged every one or two days.

### 2.4. Determination of NK-92MI Cell Activity

#### 2.4.1. Effects of OPs on NK-92MI Cell Proliferation

The NK-92MI cells were inoculated in a 96-well plate and cultured for 4 h (37 °C) in the incubator. Ten OP solutions with different concentrations of 62.5, 125, 250, 500, and 1000 μg/mL were added as the sample groups, respectively. Equal volumes (5 μL) of PBS were added as the control group. All groups were cultured for 16 h at 37 °C. The NK-92MI cell proliferation (%) was detected using the method of CCK-8.

#### 2.4.2. Effect of OPs on NK-92MI Cell Cytotoxicity

According to the modified method of Surayot and You [24], the NK-92MI cells were inoculated in a six-well plate, cultured for 4 h at 37 °C, and then separated into sample and control groups. The sample groups contained ten OP solutions with a concentration of 250 μg/mL, respectively. The control group had equal volumes (50 μL) of PBS. Both groups were cultivated for 16 h at 37 °C.

On the other hand, the Calu-1 cells were cultivated in a 96-well plate and then divided into sample, control, and blank groups. After Calu-1 cell (target cell) adhesion, the NK-92MI cells (effector cells) treated above were added with effector:target cell ratios (E:T) of 10:1. After co-cultivation for 4 h, the effect of OPs on NK-92MI cell cytotoxicity was evaluated by calculating Calu-1 cell viability according to an CCK-8 assay.

#### 2.4.3. Effect of OPs on Gene Expression of NK-92MI Cell

In accordance with the manufacturer’s protocol, the total RNA of NK-92MI cells treated in part 2.4.2 was extracted with TRIzol reagent. HiScript II QRT SuperMix for qPCR (+gDNA wiper) was used to construct cDNA. ChamQ Universal SYBR qPCR Master Mix was used for quantitative real-time PCR (qRT-PCR) amplification. The conditions of fluorescence quantitative PCR were as follows: pre-denaturation at 95 °C for 3 min and 40 cycles of 95 °C for 15 s, 57 °C for 15 s, and 72 °C for 20 s. The target gene expression of perforin (PFP), granzyme B (GZMB), interferon-γ (IFN-γ) in NK-92MI cells was determined by the fold change [2^(−ΔΔCt)^] method [25]. For qRT-PCR analysis, GAPDH was selected as an internal reference gene. The primer sequences of genes are shown in Table 1.

### 2.5. Characteristics Analysis of Polysaccharide

#### 2.5.1. Monosaccharide Composition

According to the modified method of Liu et al. [26], the OP5 was hydrolyzed, and the monosaccharides were released. Mannose, ribose, rhamnose, glucuronic acid, galacturonic acid, glucose, galactose, xylose, arabinose, fucose, *N*-acetyl glucosamine and *N*-acetyl galactosamine were dissolved in distilled water as the mixed standard solution. The monosaccharide composition of OPs was detected using an LC-20AD HPLC system with an Xtimate C18 column (Shimadzu Global Laboratory Consumables Co., Shanghai, China). The column temperature was 30 °C. The flow rate was 1.0 mL/min. The detection wavelength was 250 nm. The injection amount was 20 μL. The mobile phase was a 0.05 M mixed solution that included potassium dihydrogen phosphate and acetonitrile (volume ratio = 83:17).

#### 2.5.2. Fourier Transform-Infrared Spectroscopy (FT-IR) and Nuclear Magnetic Resonance (NMR) Analysis

After grinding and pressing, the mixture of OP5 and KBr was analyzed from 400 cm^−1^ to 4000 cm^−1^ with an Invenio R infrared spectrometer (Bruker Co., Rheinstetten, Germany). ^13^C NMR (150 MHz) and ^1^H NMR (600 MHz) spectra of the OP were recorded on a Bruker Avance III HD 600 spectrometer (Bruker Co., Rheinstetten, Germany). Deuterium oxide (D_2_O) was the solvent.

### 2.6. Statistical Analysis

All experiments were performed in triplicate. The data are presented as mean ± standard deviation. Statistically significant differences (*p* < 0.05) were determined by variance analysis (ANOVA) and SPSS 17.0 statistical software (SPSS Inc., Chicago, IL, USA).

## 3. Results and Discussion

### 3.1. Screening of Polysaccharide with the Highest Ability to Activate NK Cell

Undoubtedly, polysaccharides are substances with strong biological activity, possessing immune regulation and anti-tumor properties. Their biological features make them very favorable in functional foods and pharmaceutical fields. Screening polysaccharides with immune-stimulating activity from plants is an economical, feasible, and safe approach. As shown in Table 2, ten OPs with different molecular weights were prepared to search for the polysaccharide with the highest ability to activate NK cells by boiling water extraction, ultrafiltration-membrane separation, and ethanol precipitation. Among them, OP10 had the highest proportion with 48.41%, followed by OP8 (10.25%), OP6 (9.64%) and OP4 (8.82%). The proportion of other components was less than 7%.

The proliferation rate of NK-92MI cells co-cultured with polysaccharides is one of the factors that measure whether the polysaccharides have the ability to activate NK-92MI cells; at the very least, it can indicate at what concentration the polysaccharides do not cause damage to NK-92MI cells. It is a necessary component of polysaccharide screening work. From Figure 1, overall, we can note that the proliferation rate of NK-92MI cells treated with different concentrations of OPs was significantly higher than that of the control. It was revealed that all the OPs could promote the proliferation. With the concentration increased, the effect of every OP on NK-92MI cell proliferation showed a trend of first increasing and then decreasing. At many concentrations, the proliferation rate of NK-92MI cells exceeded 150%. For example, the proliferation rate of NK-92MI cells with the treatment of 500 μg/mL OP1, 250 μg/mL OP2, 250 μg/mL OP4, 500 μg/mL OP4, 500 μg/mL OP6, 1000 μg/mL OP6, 250 μg/mL OP8, 250 μg/mL OP9 and 500 μg/mL OP9 surpassed 150%, respectively. Among them, the proliferation rate of NK-92MI cells treated with 250 μg/mL OP9 was the highest and reached 160.40%. The higher promotion effect of each OP was displayed in the concentration from 250 μg/mL to 500 μg/mL, compared with the other concentrations of OPs. Therefore, in further experiments, the concentration range was narrowed down in order to screen out the optimal concentration.

The up-regulation of killing factors and cytokines expression is one of the remarkable characteristics of NK-92MI cells that have been activated. According to Figure 2A, the influences of different OPs on the expression of perforin in NK-92MI cells were different. At the experimental concentrations, the expression of perforin in NK-92MI cells was down-regulated by four OPs (OP7, OP8, OP9 and OP10) and up-regulated significantly by three OPs (OP2, OP4 and OP5). The level of up-regulation was relatively greater between 150 μg/mL and 250 μg/mL. As seen in Figure 2B, OP2 and OP5 could increase the expression of granzyme B in NK-92MI cells, especially in the range of 250–350 μg/mL. It was improved 1.29–1.50 fold (OP2) and 1.32–1.41 fold (OP5). As displayed in Figure 2C, the expression of IFN-γ in NK-92MI cells was enhanced by six OPs (From OP1 to OP6), respectively. All of them except OP3 resulted in a maximum up-regulation level in concentrations between 150 μg/mL and 250 μg/mL. Compared to the control, the expression level of IFN-γ was increased the most by OP5, by 5.39–7.40 fold. Based on the above experimental results, the OP concentration of 250 μg/mL was chosen for subsequent experiments.

Calu-1 cell viability can intuitively reflect the strength of NK-92MI cell cytotoxicity. In other words, it refers to the degree of killing or inhibition of Calu-1 cells by activated NK-92MI cells. When activated NK-92MI cells are co-cultured with Calu-1 cells, the stronger the cytotoxicity of NK-92MI cell is, the weaker the viability of the Calu-1 cell will be. The cytotoxicity of NK-92MI cells treated with OPs had changed, as shown in Figure 3. Specifically, the cytotoxicity of NK-92MI cells co-culturing with OP7, OP8, OP9 and OP10 decreased with varying degrees, respectively. This suggested that the NK-92MI cells activated by these OPs might not be able to release sufficient killing factors and cytokines. It was not beneficial to directly kill or collaborate with other immune cells to kill the Calu-1 cells. This result was consistent with the gene expression results of perforin, granzyme B and IFN-γ shown in Figure 2A–C. Although the other six OPs could enhance the cytotoxicity of NK-92MI cells, only OP5 reached a significant level (*p* < 0.05). Calu-1 cell viability was inhibited to 83.94% by NK-92MI cells treated with OP5 at a concentration of 250 μg/mL. Under the same conditions, the expression levels of perforin and granzyme B, the main killer factors of NK-92MI cells, were up-regulated by 4.07-fold and 1.32-fold, respectively. This indicated that OP5 was likely to inhibit the vitality of Calu-1 cells by stimulating NK-92MI cells to increase the secretion of killer factors. In summary, based on the experimental results of cell proliferation, cytokine secretion, and cytotoxicity, the capacity of OP5 for activating NK-92MI cells was, relatively, the highest.

In general, the OPs with different molecular weights had different capabilities to activate NK-92MI cells. Many studies have proved that appropriate molecular weight is the foundation for polysaccharides to exert their strongest biological activity [27,28]. According to Table 3 in this study, under the condition of 250 μg/mL, the difference in molecular weight did not significantly affect ability of OPs to promote NK-92MI cell proliferation. However, polysaccharides with a molecular weight between 3 kDa and 100 kDa (OP1-OP6) possessed a stronger ability to up-regulate the expression of perforin and IFN-γ in NK-92MI cells and inhibit Calu-1 cell viability than polysaccharides with molecular weights >100 kDa (OP7-OP10). It seemed that the polysaccharides with a molecular weight less than 100 kDa were more conducive to their immuno-stimulating activity. This inference was consistent with the results of some other researchers [29,30,31,32,33,34,35]. For instance, Huo et al. [29] extracted a novel polysaccharide, HSP-2, with a molecular weight of 39.8 kDa, which showed significant immuno-stimulating effects by increasing the ROS and NO generation of THP-1 cells in a dose-dependent manner. Mirzaie et al. [30] obtained three polysaccharides—CP (161.1 kDa), F1 (23.9 kDa) and F2 (41.9 kDa)—from *Chlorella vulgaris*. The up-regulation of IFN-γand IL-2 in chicken peripheral blood mononuclear cells incubated with F1 was greater than CP and F2. Zhou et al. [33] pointed out that the molecular weight of polysaccharides had a notable effect on immunomodulation activities. Among five polysaccharides prepared from *Chondrus ocellatus* (PC1 652 kDa, PC2 238 kDa, PC3 143 kDa, PC4 65 kDa and PC5 9.3 kDa), the antitumor activity of PC5 on an S180 tumor was the highest, while the antitumor activity of PC4 on an H22 tumor was the highest.

### 3.2. Characteristics of OP5

Based on the research conclusion in Section 3.1, further study on the characteristics of OP5 has been carried out. As seen in Figure 4, in comparison with the HPLC standards, it was found that OP5 was composed of 10 monosaccharides. Their proportions were arabinose: 31.52%, galacturonic acid: 22.35%, galactose: 16.72%, glucose: 15.95%, mannose: 7.67%, rhamnose: 2.39%, fucose: 1.41%, xylose: 1.30%, glucuronic acid: 0.42% and ribose: 0.27%. The high content of galacturonic acid indicated that OP5 was an acidic polysaccharide.

From Figure 5, we can observe that the strong absorption peak at 3416 cm^−1^ was caused by the stretching vibration of the O-H bond. The absorption peak at 2936 cm^−1^ was attributed to the stretching vibration of the C-H bond [36]. Two absorption peaks at 1751 cm^−1^ and 1609 cm^−1^ should be the symmetrical and asymmetrical stretching vibration peaks of the carboxyl (-COOH) C=O bond, respectively [37,38], which revealed the existence of glycuronate in OP5. An absorption peak at 1443 cm^−1^ is due to the bending vibration of the C-H bond [39]. The absorption peaks at 1234 cm^−1^ and 1105 cm^−1^ might result from the stretching vibration of the C-O bond. A strong absorption peak was observed at 1018 cm^−1^, suggesting that there were pyranose ring and stretching vibration of the C-O-C bond in OP5. The absorption peak at 633 cm^−1^ may be caused by the out-of-plane bending vibration of the O-H bond.

According to Figure 6A, in the anomeric proton region (δ 4.3–5.9), there was an obvious signal at δ 4.70 that was less than 5.0, mainly implying that OP5 had the β-configuration of a sugar ring. The signal at δ 1.26 should be attributed to the protons of methyl in rhamnose. In Figure 6B, in the characteristic signal region of glycuronate (δ 170–190), the clear signal at δ 170.66 showed that OP5 had the glycuronate. This result was consistent with the detection result of monosaccharide composition. Within the anomeric carbon region (δ 90–112), many signals appeared in the low field (δ 102–112) and several signals appeared in the high field (δ 90–102), indicating that the sugar-ring was mainly of a β-configuration, and the α-configuration of the sugar ring still existed. Two signals at δ 82.33 and δ 83.88 declared the possible presence of a furanose structure in OP5. In addition, four signals (δ 81.25, 80.66, 79.01 and 76.53) located in the region from δ 76 to δ 85 and three signals (δ 74.42, 74.12 and 71.11) located in the region from δ 70 to δ 75 illustrated that the C2, C3 and C4 carbons of pyranose had been partially replaced. The signals at δ 69.65, δ 69.18 and δ 67.53 showed that C6 on some pyranoses was substituted. There were four signals (δ 63.86, 63.34, 61.07 and 60.73) that occurred between δ 60 and δ 64, implying that C6 on some pyranoses had not been replaced. The signal at δ 52.86 should be attributable to the carbon of the *N*-substituted group in *N*-Ac (CH_3_CON-). The presence of a signal at δ 23.85 revealed that it might be caused by the carbon of methyl in *N*-Ac. A signal at δ 19.89 possibly resulted from the carbon of methyl in O-Ac (CH_3_COO-).

Some special structural characteristics of polysaccharides are beneficial for activating NK cells, which has been confirmed by many studies [40,41]. For instance, polysaccharides with glucose [24,42], mannose [43,44] or *N*-acetyl-d-glucosamine [24] are more easily recognized and connected by complement receptor type 3 (CR3) on NK cells [45]. Vetvicka et al. [46] reported that a soluble polysaccharide, SZP (10 kDa, 95% mannose and 5% glucose), had a higher affinity for CR3 than many soluble β-glucans (20–1000 kDa). This indicated that polysaccharides rich in mannose had more advantages than polysaccharides rich in glucose when binding with CR3 to stimulate NK cells to release cytokines and kill mediators, thereby eliminating tumor cells or pathogen-infected cells. In this investigation, OP5 possessed 22.35% galacturonic acid. It was beneficial for polysaccharides to dissolve quickly in water and exert subsequent immune effects [41,47,48,49,50]. OP5 also contained 15.95% glucose and 7.67% mannose. This was conducive to binding OP5 to CR3 in NK-92MI cells. Then, NK-92MI cells were stimulated to up-regulate the secretion of perforin and granzyme B to kill Calu-1 cells.

## 4. Conclusions

Among ten polysaccharides extracted and isolated from orah mandarin peel, OP5 has been screened out because of its immunostimulatory activity on NK-92MI cells. Under the condition of 250 μg/mL, the proliferation rate was promoted (145.09%), the expression of genes (perforin 4.07 fold, granzyme B 1.32 fold, IFN-γ 7.40 fold) was up-regulated, and Calu-1 cell viability was inhibited (83.94%) by OP5. OP5 was an acidic polysaccharide composed of arabinose (31.52%), galacturonic acid (22.35%), galactose (16.72%), glucose (15.95%), mannose (7.67%), rhamnose (2.39%), fucose (1.41%), xylose (1.30%), glucuronic acid (0.42%), and ribose (0.27%). Its molecular weight was between 50 kDa and 70 kDa. The strongest ability of OP5 to activate NK cells among the ten OPs might be related to its moderate molecular weight and high content of galacturonic acid, glucose, and mannose. However, it is still unknown how the exact structure of OP5 affects NK-92MI cell activity. Therefore, OP5 needs to be purified to determine its exact structure, such as glycosidic linkage type and chain conformation. Experiments on how the exact structure of polysaccharides binds to NK-92MI cell receptors and regulates the expression of genes associated with these proteins in pathways will also be carried out in the future. The research results will be beneficial for the functionalized development and utilization of orah mandarin.

## Figures and Tables

**Figure 1 foods-13-00082-f001:**
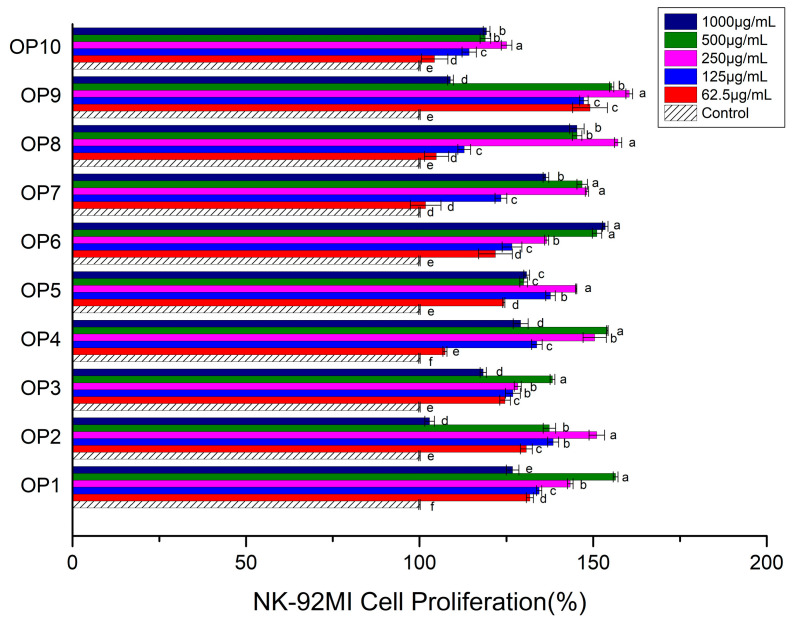
Effects of different OPs on NK-92MI cell proliferation. The values are presented as mean ± SD. Different letters indicate significant differences (*p* < 0.05) between the data within each group.

**Figure 2 foods-13-00082-f002:**
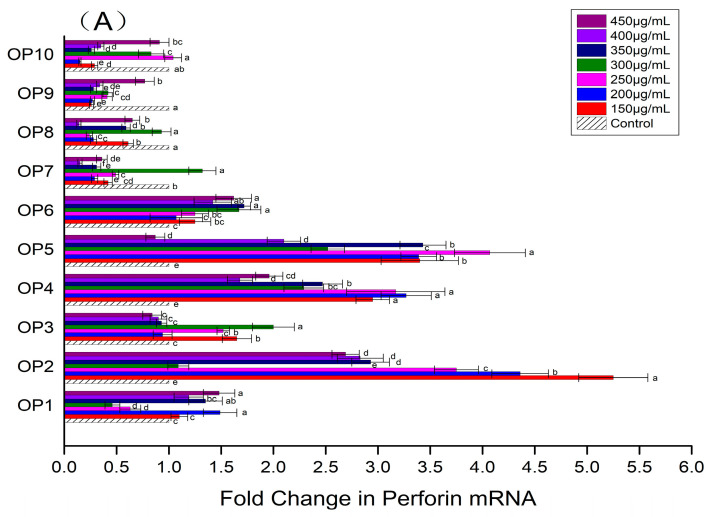
Effects of different OPs on expression level of perforin (**A**); expression level of granzyme B (**B**); expression level of IFN-γ (**C**) in NK-92MI cell. The values are presented as mean ± SD. Different letters indicate significant differences (*p* < 0.05) between the data within each group.

**Figure 3 foods-13-00082-f003:**
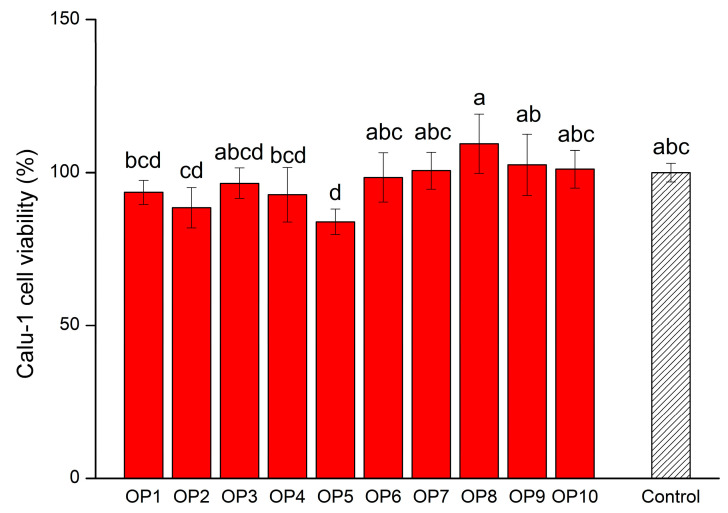
Effects of different OPs on NK-92MI cell cytotoxicity. The values are presented as mean ± SD. Different letters indicate significant differences (*p* < 0.05) between the data within each group.

**Figure 4 foods-13-00082-f004:**
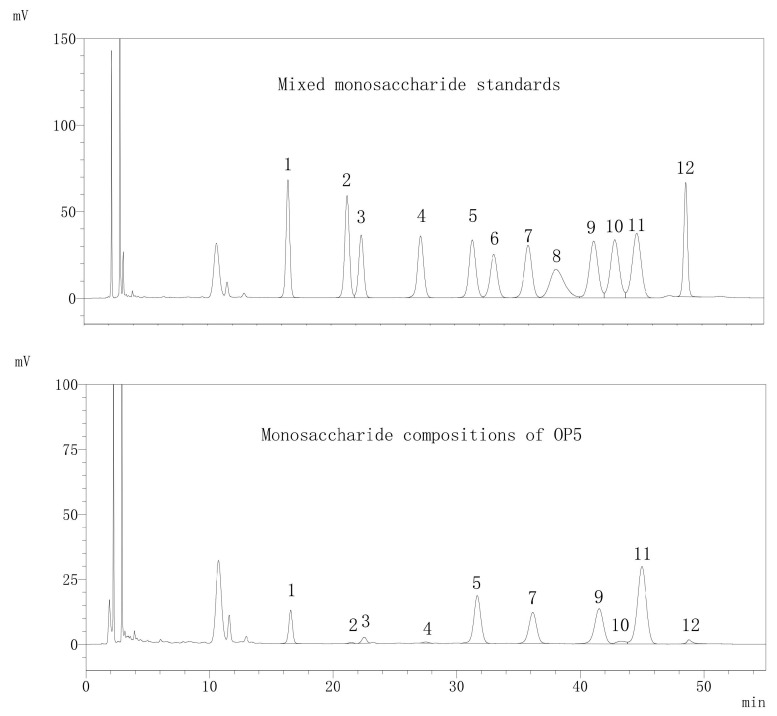
HPLC of monosaccharides of OP5 (1: mannose, 2: ribose, 3: rhamnose, 4: glucuronic acid, 5: galacturonic acid, 6: *N*-acetyl glucosamine, 7: glucose, 8: *N*-acetyl galactosamine, 9: galactose, 10: xylose, 11: arabinose, and 12: fucose).

**Figure 5 foods-13-00082-f005:**
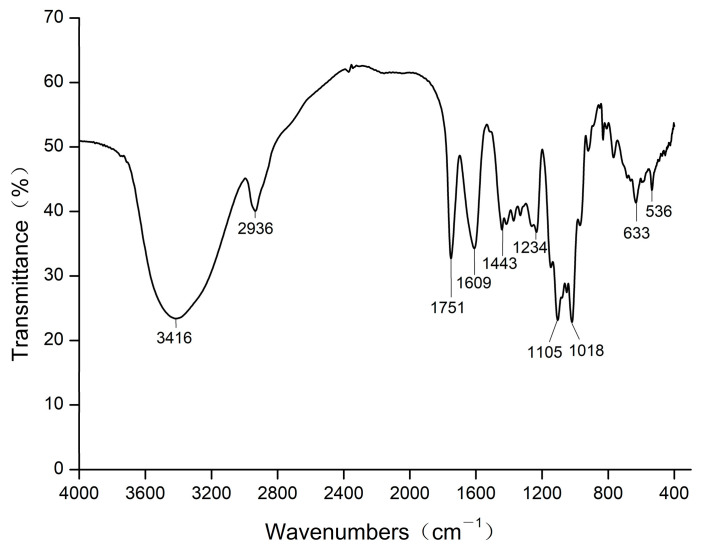
Infrared spectrum of OP5.

**Figure 6 foods-13-00082-f006:**
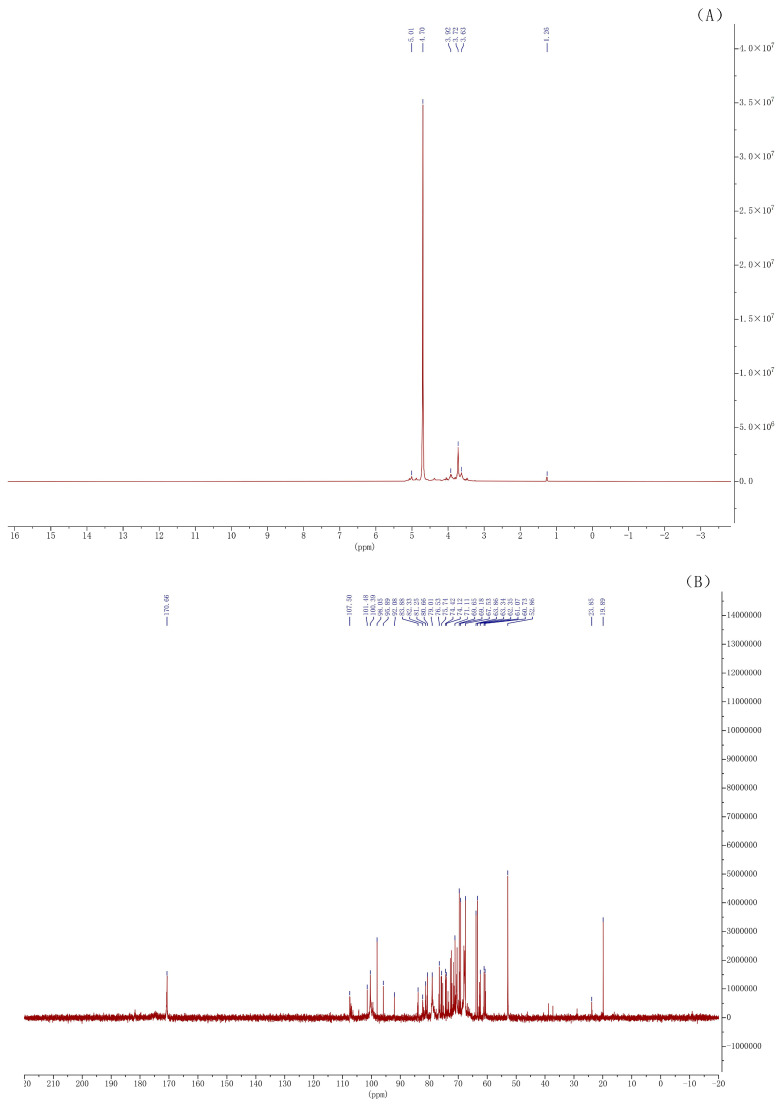
^1^H NMR spectrum (**A**); and ^13^C NMR spectrum (**B**) of OP5.

**Table 1 foods-13-00082-t001:** Primer sequences of real- time PCR analysis.

Gene	Sequences of the Primers	Product Size (bp)
GAPDH	Forward: GAGTCCACTGGCGTCTTCAC Reverse: TGCTGATGATCTTGAGGCTGTT	157
Perforin	Forward: GACTGCCTGACTGTCGAGG Reverse: TCCCGGTAGGTTTGGTGGAA	128
Granzyme B	Forward: CCCTGGGAAAACACTCACACA Reverse: GCACAACTCAATGGTACTGTCG	110
IFN-γ	Forward: TCGGTAACTGACTTGAATGTCCA Reverse: TCGCTTCCCTGTTTTAGCTGC	93

**Table 2 foods-13-00082-t002:** Different molecular weights and proportions of OPs.

Molecular Weight (kDa)	Named as	Proportion (%)
3–5	OP1	1.05 ± 0.08
5–10	OP2	2.56 ± 0.19
10–30	OP3	3.38 ± 0.21
30–50	OP4	8.82 ± 0.50
50–70	OP5	3.93 ± 0.29
70–100	OP6	9.64 ± 0.56
100–300	OP7	6.55 ± 0.43
300–500	OP8	10.25 ± 0.87
500–1000	OP9	5.41 ± 0.38
>1000	OP10	48.41 ± 1.88
Total		100.00

**Table 3 foods-13-00082-t003:** Effects of 10 OPs on NK cell activity under the condition of 250 μg/mL.

Polysaccharide	NK-92MI Cell Activity				
	Proliferation (%)	Perforin	Granzyme B	IFN-γ	Calu-1 Cell Viability (%)
OP1 (3–5 kDa)	143.36 ± 0.81	0.63 ± 0.10	0.85 ± 0.08	5.19 ± 0.20	93.55 ± 4.00
OP2 (5–10 kDa)	151.04 ± 2.23	3.75 ± 0.21	1.50 ± 0.10	3.41 ± 0.35	88.54 ± 6.60
OP3 (10–30 kDa)	128.25 ± 1.00	1.52 ± 0.06	0.68 ± 0.07	3.91 ± 0.20	96.53 ± 4.98
OP4 (30–50 kDa)	150.44 ± 3.36	3.17 ± 0.47	0.58 ± 0.03	5.80 ± 0.27	92.78 ± 8.88
OP5 (50–70 kDa)	145.09 ± 0.17	4.07 ± 0.34	1.32 ± 0.08	7.40 ± 0.33	83.94 ± 4.14
OP6 (70–100 kDa)	136.56 ± 0.59	1.25 ± 0.13	0.88 ± 0.07	2.51 ± 0.20	98.42 ± 8.05
OP7 (100–300 kDa)	148.20 ± 0.48	0.49 ± 0.03	0.80 ± 0.07	1.51 ± 0.15	100.63 ± 5.98
OP8 (300–500 kDa)	157.16 ± 1.05	0.24 ± 0.03	0.60 ± 0.07	0.84 ± 0.08	109.45 ± 9.68
OP9 (500–1000 kDa)	160.40 ± 0.96	0.41 ± 0.05	0.85 ± 0.09	1.07 ± 0.03	102.55 ± 10.02
OP10 (>1000 kDa)	125.03 ± 1.52	1.04 ± 0.08	0.99 ± 0.08	0.97 ± 0.09	101.11 ± 6.11

## Data Availability

Data is contained within the article.

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
