# Peer review of "Screening and Characteristics Analysis of Polysaccharides from Orah Mandarin (Citrus reticulata cv. Orah)"

_foods, 2023, doi:10.3390/foods13010082_

Round 1
Reviewer 1 Report
Comments and Suggestions for Authors
Reviewer comments:05.12.2023
A manuscript with the title; “Screening and characteristics analysis of polysaccharide from orah mandarin (Citrus reticulata cv. Orah)” by Liu et al. screen out the polysaccharides having ability to activate NK cell. The experiments on how exact structure of polysaccharide binds to NK-92MI cell receptors and regulates expression of genes associated with these proteins in pathways will also be carried out in the future. The research results will be beneficial for the functionalized development and utilization of.
General conclusions, suggestions for the authors:
1. The manuscript is written in correct English.
2. Please adapt the structure of the manuscript to the requirements of the journal in accordance with the instructions for the author.
3. The methods used in this work are selected adequately to the purpose of the research.
4. The work shows potential for further research on this issue
Below my comments to the authors:
1. Line 18. The word “cyto – toxicty” is divided by a dash and moved to a new line. This is not an error, but it disturbs the reading of the text and does not look aesthetically pleasing. There are many transfers of this type in the manuscript. If it is possible, please correct it for better reception of the manuscript. “cyto – toxicty” should be “cytotoxicty”.
2. Line 90. Please provide the initial amount of orah mandarin peel before and after drying.
3. Line 92. Please provide the grinding speed.
4. Line 99. Please enter the amount of orah mandarin peel dissolved in boiling water.
5. Line 147. Table 1 has too large a font and should be changed.
6. Line 182. Table 2 has too large a font and should be changed.
7. Line 192. Fig. 1 is not high quality, it should be higher resolution. Maybe it would be better to use one of the figures and put the others in Supplementary. The description of the remaining ones should be included in the Results chapter.
8. Line 233. Table 3 has too large a font and should be changed.
9. Line 249. Similar situation like in Fig. 1. This fig is better to put in Supplementary.

Author Response
Thank you very much for your advise on our manuscript. We have endeavored to address all of the comments, and the amendments are highlighted in red in the revised manuscript. These comments were helpful to improve greatly the manuscript. We trust that the revised version of the manuscript meets the requirements of the journal. Thank you so much for your time and supporting our efforts in the improvement of the manuscript.
Replying Reviewer 1 comments:
- Line 18. The word “cyto – toxicty” is divided by a dash and moved to a new line. This is not an error, but it disturbs the reading of the text and does not look aesthetically pleasing. There are many transfers of this type in the manuscript. If it is possible, please correct it for better reception of the manuscript. “cyto – toxicty” should be “cytotoxicty”.
Author’s response: We have made corresponding modifications in the revised manuscript based on the suggestion. For example, the words “cycle-toxicity”, “signaling-cantly” and “β-configuration-tion”have been modified to “cycletoxicity”, “signalingcantly” and “β-configurationtion”.
- Line 90. Please provide the initial amount of orah mandarin peel before and after drying.
Author’s response: The initial amount of orah mandarin peel before and after drying have been complemented in line 99 and 100.
- Line 92. Please provide the grinding speed.
Author’s response: The speed of FW 177 high rotated speed disintegrator has been complemented in line 102.
- Line 99. Please enter the amount of orah mandarin peel dissolved in boiling water.
Author’s response: The amount of orah mandarin peel polysaccharide dissolved in boiling water has been complemented in line109.
- Line 147. Table 1 has too large a font and should be changed.
Author’s response: The font of Table 1 has been reduced according to the suggestion.
- Line 182. Table 2 has too large a font and should be changed.
Author’s response: The font of Table 2 has been reduced according to the suggestion.
- Line 192. Fig. 1 is not high quality, it should be higher resolution. Maybe it would be better to use one of the figures and put the others in Supplementary. The description of the remaining ones should be included in the Results chapter.
Author’s response: According to the suggestion, we have doubled the resolution of all images and also split Figure 1 into Figure 1 (Line215-218), Figure 2 (234-239) and Figure 3 (260-263).
- Line 233. Table 3 has too large a font and should be changed.
Author’s response: The font of Table 3 has been reduced according to the suggestion.
- Line 249. Similar situation like in Fig. 1. This fig is better to put in Supplementary.
Author’s response: According to the suggestion, we have doubled the resolution of all images and split Figure 2 into Figure 4 (Line295-298), Figure 5 (310-311) and Figure 6 (331-333).

Reviewer 2 Report
Comments and Suggestions for Authors
The manuscript provides clear detailing of the isolation of polysaccharides from orah mandarin peel and their effects on NK-92MI cell proliferation and cytotoxicity.
Some considerations should be taken into account by the authors:
Introduction
To enhance the clarity and flow of the introduction, the transition between the general information about citrus peel and the specific focus on NK cell activation could be made more seamless. Additionally, providing a brief preview of the methodology or key outcomes in the introduction could engage the reader and offer a roadmap for what to expect in the study.
section 3.1:
- Consider briefly discussing any notable observations in the proliferation rates, especially in the context of concentration-dependent effects.
- Elaborate on the significance of the observed decrease in cytotoxicity and its potential implications for the overall immune response.
- Discuss any existing literature or theories that support the idea that polysaccharides with molecular weights less than 100 kDa may be more conducive to immunostimulating activity. Address any consistencies or differences in the observed immunostimulating activity in relation to molecular weight.
Section 3.2.
Consider drawing comparisons between the characteristics of OP5 and known polysaccharides with immune-stimulating properties. Highlight any unique features of OP5 and discuss their potential significance in the context of its application as a functional compound.
Discuss how the identified structural features of OP5 may contribute to its observed effects on NK cell activation. Link the polysaccharide's composition and structure to potential mechanisms of action, drawing connections with established immunological pathways.
Author Response
Thank you very much for your advise on our manuscript. We have endeavored to address all of the comments, and the amendments are highlighted in red in the revised manuscript. These comments were helpful to improve greatly the manuscript. We trust that the revised version of the manuscript meets the requirements of the journal. Thank you so much for your time and supporting our efforts in the improvement of the manuscript.
Replying Reviewer 2 comments:
- Introduction: To enhance the clarity and flow of the introduction, the transition between the general information about citrus peel and the specific focus on NK cell activation could be made more seamless. Additionally, providing a brief preview of the methodology or key outcomes in the introduction could engage the reader and offer a roadmap for what to expect in the study.
Author’s response: We have made appropriate modifications in Section 1 (Line 50-55, Line 75-81) based on the suggestion .
- 2. Section 3.1:
- Consider briefly discussing any notable observations in the proliferation rates, especially in the context of concentration-dependent effects.
- Elaborate on the significance of the observed decrease in cytotoxicity and its potential implications for the overall immune response.
- Discuss any existing literature or theories that support the idea that polysaccharides with molecular weights less than 100 kDa may be more conducive to immunostimulating activity. Address any consistencies or differences in the observed immunostimulating activity in relation to molecular weight.
Author’s response: According to the suggestion, we have added the discussion about effect of polysaccharides on NK cell proliferation rate (Line 198-211). The possible reasons and effects of decrease in cytotoxicity have been supplemented (Line 240-250). The discussion about supporting the idea that polysaccharides with molecular weights less than 100 kDa may be more conducive to immunostimulating activity has been supplemented (Line 274-285).
- 3. Section 3.2:
Consider drawing comparisons between the characteristics of OP5 and known polysaccharides with immune-stimulating properties. Highlight any unique features of OP5 and discuss their potential significance in the context of its application as a functional compound.
Discuss how the identified structural features of OP5 may contribute to its observed effects on NK cell activation. Link the polysaccharide's composition and structure to potential mechanisms of action, drawing connections with established immunological pathways.
Author’s response: According to the suggestion, we have made appropriate modifications and additions (Line 334-347).
